# Current Situations and Challenges in the Development of Health Information Literacy

**DOI:** 10.3390/ijerph20032706

**Published:** 2023-02-02

**Authors:** Qiulin Wang, Chunhua Tao, Yuan Yuan, Song Zhang, Jingyan Liang

**Affiliations:** 1College of Physical Education, Yangzhou University, Yangzhou 225009, China; 2School of Nursing and School of Public Health, Yangzhou University, Yangzhou 225009, China; 3School of Medicine, Yangzhou University, Yangzhou 225009, China; 4Institute of Translational Medicine, Medical College, Yangzhou University, Yangzhou 225009, China; 5Jiangsu Key Laboratory of Integrated Traditional Chinese and Western Medicine for Prevention and Treatment of Senile Diseases, Yangzhou University, Yangzhou 225009, China

**Keywords:** health information literacy, assessment tools, influencing factors, health information sources, doctor–patient relationship

## Abstract

Health information literacy (HIL) is a significant concept that has gradually become known to the broader public in recent years. Although the definitions of HIL and health literacy seem to overlap, as an independent subconcept, HIL still shows a unique influence on improvements in people’s health and health education. Remarkable evidence indicates that online health information (OHI) can effectively enrich people’s knowledge and encourage patients to actively join the medical process, which is also accompanied by the emergence of various assessment tools. Although the current assessment tools, to a certain extent, can help people identify their shortcomings and improve their HIL, many studies have indicated that the deficiencies of the scales induce incomplete or unreal results of their HIL. In addition, continuing research has revealed an increasing number of influencing factors that have great effects on HIL and even regulate the different trends in doctor–patient relationships. Simultaneously, most of the uncensored OHI broadcasts have also affected the improvement in HIL in various ways. Thus, this review aims to summarize the assessment tools, influencing factors and current situations and challenges related to HIL. Further studies are required to provide more trusted and deeper references for the development of HIL.

## 1. Background

Since the term health literacy was first coined in 1974 by Simonds SK, an increasing number of studies have been devoted to exploring the interaction between health literacy and public health [1]. Many studies have considered that it could affect medical outcomes. In a systematic review including 31 publications, the researchers reported that the definition of health literacy was various in different papers, and they tried to summarize and redefine health literacy, which was described as ‘Health literacy is linked to literacy and entails people’s knowledge, motivation and competences to access, understand, appraise, and apply health information in order to make judgments and take decisions in everyday life concerning healthcare, disease prevention and health promotion to maintain or improve quality of life during the life course’ [2]. Additionally, it was also mainly divided into three different categories requiring incremental levels of knowledge and skills, functional health literacy, interactive health literacy and critical health literacy, in the prototypical model of Nutbeam [2,3]. Along with rapid health literacy development, a new subconcept—health information literacy (HIL)—took shape and was finally proposed in 2003 by the Medical Library Association (MLA), combining information and health literacy [4]. In contrast to health literacy, the emphasis of HIL is that humans play a subject role instead of an object role when carrying out information discovery [5]. Thus, HIL was defined as people recognizing the need for health information, knowing how and where to find information about health, and knowing how to evaluate and use such information to make good health decisions [6,7]. When people improve these abilities, they can actively advance their health transition [8].

In addition, HIL is a vital component of health literacy and shows unique effects on personal health literacy, which is considered a panacea for poverty alleviation in developing countries [9]. Simultaneously, some developed countries have also considered that enhancing HIL is a priority [10]. However, improving the HIL of the population is still very difficult. The currently existing assessment tools are criticized for not truly reflecting the HIL level of the population [6]. Except for HIL-related abilities, many influencing factors, such as sociodemographic and disease factors, still affect people’s attitudes and motivations in the process of improving their HIL. Additionally, incorrect and uncensored online health information (OHI) has become a major problem that inhibits the development of HIL. Many studies have demonstrated the existence of potential harmful topics, such as cancer incidence, fever management in children and smoking cessation methods, by comparing the clinical evidence with OHI [11,12,13]. Last but not least, as the main role in the doctor–patient relationship and health education, the medical staff should not be neglected. Obviously, people’s HIL levels can influence the interaction between medical staff and patients [14].

Therefore, this review attempts to integrate the existing relevant literature and make conclusions about the current development of HIL in a relatively comprehensive way by summarizing the assessment tools, influencing factors, and current situations and challenges related to HIL to provide a reference for establishing more effective management strategies to improve people’s HIL.

## 2. Assessment Tools

To our knowledge, an individual’s level of HIL plays a significant role in the results of health education and has a profound impact on tertiary prevention. Realizing the importance of HIL, an increasing number of scholars have made an effort to develop a suitable assessment instrument. Based on the theoretical conceptualization of HIL, present measurement tools mainly focus on the following dimensions: screening reading ability, motivation to find health information, information-seeking ability, evaluating the quality of health-related materials, and understanding and applying health information. The characteristics of the tools used to assess HIL are summarized, and their strengths and limitations are listed in Table 1.

### 2.1. Rapid Estimate of Adult Literacy in Medicine (REALM)

The Rapid Estimate of Adult Literacy in Medicine (REALM), established by Davis et al. in 1991 [15], is a popular instrument that is widely applied to measure the ability to read and spell common medical terms selected from printed patient education materials. Participants are asked to read a list of 66 health-related words that become progressively difficult. An individual’s REALM score is the total number of correctly pronounced words without arbitrarily adding or removing the beginning or end of a word [16]. The REALM mainly aims to identify the individual’s reading ability and is suitable for people with limited literacy.

### 2.2. DISCERN Scale

In 1995, Charnock et al. [17] developed a short scale called DISCERN with the cooperation of the British Library and the University of Oxford to assess the quality of treatment-related information. The DISCERN scale is divided into two dimensions: evaluating the reliability of a website (consisting of 8 items and a total score of 40) and verifying the quality of articles (including 7 items and a total score of 35). Despite the subjective format of the DISCERN scale, it is valid and can be applied to healthcare professionals and patients [18].

### 2.3. Functional Health Literacy Test in Adults (TOFHLA)

Reading, writing and computational skills are critical and basic factors for the successful understanding and application of health-related materials. Parker et al. [19] developed an instrument on the fundamentals of actual medical documents to better measure functional health literacy in 1995. Functional health literacy means the individual’s ability to apply literacy skills (i.e., basic reading, writing and computational skills) to health-related materials, which is a significant component of HIL. The TOFHLA consists of two parts, including a 17-item test for numeracy and a 50-item test for reading comprehension. Despite the limited evaluation of several HIL dimensions, the TOFHLA can provide directions in primary healthcare settings [20].

Given that the TOFHLA takes 22 min to complete, Baker et al. [21] revised the original TOFHLA and revised it by reducing 17 numeracy items and 50 reading comprehension items to create a short version of 4 numeracy items and 36 reading comprehension items. The maximum time needed to complete the short TOFHLA (STOFHLA) was reduced from 22 to 12 min. Notably, neither the original nor the short version of the TOFHLA is a valid and available instrument.

### 2.4. Newest Vital Sign (NVS)

To overcome the difficulty that current instruments that screen health information literacy are too long for routine use, in 2005, Weiss et al. [22] established a quick and accurate instrument for limited literacy called the Newest Vital Sign (NVS). The NVS mainly uses food labels to evaluate the reading comprehension and numeracy abilities of individuals. Completing the NVS takes only 3–6 min. The NVS test is suitable for the rapid assessment of low HIL.

### 2.5. Research Readiness Self-Assessment Tool-Health (RRSA-h)

Based on the Information Literacy Competency Standards for Higher Education, Ivanitskaya et al. [23] developed an instrument to measure proficiency in finding, obtaining and evaluating electronic health information in combination with information literacy conception and health literacy, namely, the Research Readiness Self-Assessment Tool-Health (RRSA-h). The RRSA-h consists of 56 entries, 16 multiple-choice questions and 40 true-or-false questions. This self-administered instrument targets college-age health information consumers and the findings can be utilized to suggest that HIL educators improve educational interventions.

### 2.6. Everyday Health Information Literacy Screening Tool (EHIL)

Based on the theoretical framework of the Medical Library Association’s (MLA) conception of health information literacy, Niemelä et al. [6] developed a short, 10-item Everyday Health Information Literacy Screening Tool (EHIL). The purpose of the EHIL is to detect individuals’ motivation and interest in finding, understanding, evaluating and using health information. To the best of our knowledge, the EHIL is the first attempt to systematically design an assessment tool that is feasible for evaluation based on a few items.

### 2.7. Chinese Version of the Health Information Literacy Self-Rating Scale (HILSS)

On the basis of the health information literacy theory proposed by the MLA, Wang et al. [24] designed a conceptual framework of the scale and screened the items of the self-rating scale, finally forming a 29-item Chinese version of the Health Information Literacy Self-Rating Scale (HILSS). The HILSS was designed to describe the individual’s health information awareness, access, evaluation, use and ethics. Despite the potential bias in adopting self-reported methods, the HILSS is a promising tool for Chinese residents in primary healthcare settings to screen for potential HIL problems.

### 2.8. Chinese Version of the Health Information Literacy Questionnaire (HILq)

Because no specialty has been applied in current health information literacy, Liu et al. [25] adopted the item pool and Delphi method based on theoretical research to establish a questionnaire to screen the health information literacy of patients (HILq) with chronic kidney disease (CKD). The instrument is divided into six dimensions, including health information acquisition, evaluation, literacy awareness, application, integration and CKD health knowledge reserve. The HILq provides significant guidance for the health education of CKD.

### 2.9. The Influencing Factors of HIL

Given the deepening of research on HIL, more research has focused on the influencing factors of HIL. On the basis of previous studies, we mainly summarized the related influencing factors and compared them with each other to analyze the potential correlation. According to the factors included, we divided them into three different types: innate demographic characteristics, acquired behavior and environment, and disease factors.

### 2.10. Innate Demographic Characteristics

At present, many studies have shown that age, sex and race are obvious factors that affect the HIL of different people [26,27,28]. Regarding age, most studies concluded that it was an essential factor in indicating the degree of HIL in various people [4,27,29,30,31]. Mao et al. [4] and Wang et al. [27] considered that older people exhibit worse HIL than young people because of the degradation in cognitive and learning abilities. Similarly, some studies have concluded that young people not only possess better HIL but are also willing to use OHI as a reference more frequently [29,30,32]. Older people may believe in professional-like medical staff rather than online information, while young people positively check various health information through multiple approaches. However, one study found no difference between the young and the old in this aspect [33].

Given the complex differences in diverse age groups, including physical and psychological changes, more conditions should be incorporated in combination with age to analyze the HIL and not just perform group analysis via age alone.

Regarding sex, some studies demonstrated that females exhibit better HIL than males at a young age [34,35] and tend to utilize OHI in daily life [36]. Regarding old adults, elderly men appear to be more highly motivated to search for health information than females [37,38]. This opposite phenomenon may exist because of certain factors, such as lifestyle, smoking, drinking and different onset times of illness between men and women, which is interesting and valuable to explore in detail in the future. Trying to discover the motivation and improve the attitudes of people to raise their HIL may be a worthwhile approach.

With respect to race, Rooks RN et al. [35] came to the conclusion that Latino individuals might search less for OHI than white individuals, whereas African American individuals, by contrast, show more interest in and motivation to use OHI. In addition, many studies also summarized the characteristics and considered that ethnic minorities might more frequently exhibit lower HIL [39,40]. The lack of better living and cultural environments and language or communication limitations may be the causative mechanism [41,42]. Thus, to raise the quality of HIL in ethnic minorities and improve their health situations, it will be very important to develop the conditions and provide convenient approaches for them to utilize healthcare-related resources.

## 3. Acquired Behavior and Environment

Education level, income and type of occupation are strongly interlinked, and many studies have evaluated their effects on HIL [34,43,44,45]. Eriksson-Backa, K et al. [31] and Sedrak MS et al. [34] showed that people with a high education level are more likely to obtain a higher HIL. In contrast, it was also found that people with relatively low education levels have low HIL [45,46,47]. In addition, people with high education levels are inclined to search for useful OHI because of their better recognition and understanding of health information [43,48]. Therefore, great significance is improving the cognitive level of residents to improve the level of HIL via education.

Income is also one of the great factors influencing residents’ HIL. Based on previous studies, we found that people with a higher income appear to have better HIL than those with a lower income, especially regarding their attitude toward and motivation to search for OHI [36,49,50]. Regarding the type of occupation, it has been concluded that careers such as medical staff possess better HIL than other jobs, while workers and peasants show relatively lower HIL [4,45,49]. In consideration of the characteristics of these factors, providing sufficient and readily available health-related resources for such populations will be valuable and worthwhile to help them improve their HIL.

Apart from the above-influencing factors, the cultural environment and experience of using the internet may also exert an influence on HIL. As a crucial and convenient medium, the internet plays a vital role in promoting OHI in people’s daily lives. Thus, several studies have concluded that people who can smoothly take advantage of the network system possess better HIL than those without such an ability [34,51,52]. In addition, an interesting phenomenon was discovered in one study in which the cultural environment also affected HIL. Yu-Chan Chiu [53] found that a hierarchical culture in society could make some patients follow doctors’ instructions unconditionally instead of learning health information and becoming involved with the medical process via various media, which deeply influences their HIL.

## 4. Disease Factors

According to the influence of disease factors on HIL, many conditions should be considered, such as the course of diseases, disease patterns and healthy propaganda. There is no doubt that the course of diseases is a significant factor affecting patients’ HIL [4]. However, it did not always show the same impact on HIL. Mao et al. [4] considered that a shorter disease course could benefit patients’ HIL, which may make them focus more on the disease, tend to search for OHI and actively communicate with the medical staff. In contrast, many patients with a long course of diseases—chronic diseases—such as diabetes and cancer, showed better HIL and appeared to have a higher demand for OHI [54,55,56]. Nevertheless, the opposite conclusions were also reached: worse HIL was evaluated in people with a long course of diseases [31,57]. This may be because patients live with their diseases for too long and accumulate some experience during this time, which makes them neglect to update their related knowledge, gradually reducing their enthusiasm. In addition, the quality and attitude of healthy propaganda from the medical staff are also very momentous during the patient’s learning process [4]. Thus, predicting the HIL only based on the course of the disease is not exact, and more conditions should be considered regarding the disease factors. On the basis of the complex correlation, more attention should be paid to the research on the effect of disease factors on patients’ HIL.

## 5. Benefits, Problems and Challenges

### 5.1. Assessment Tools

Given the development of research on HIL, an increasing number of assessment tools have been designed and used to evaluate the HIL of people, an active promotional approach to improving people’s HIL by discovering shortages and proposing improvement strategies. However, many problems have also gradually emerged through this process.

Although many studies have assessed HIL in various ways, standard methods for its evaluation are still lacking. Based on HIL theory, Wang et al. [24] designed the Chinese version of the HILSS, which contains health information cognition, search, evaluation, application and ethics. This version assesses HIL via self-appraisal, which may be easily affected by subjective factors related to the evaluation object, creating errors [4]. In view of REALM, more emphasis was placed on rapid clinical assessment, and its results might be fragmented through an evaluation of the ability to acquire information [16]. In addition, the TOFHLA focuses more on functional health information literacy [19,21]. Given the definition of HIL, which includes four main components previously mentioned [6], it is difficult for existing assessment tools to reflect HIL levels accurately. In addition, inconsistent evaluation standards lead to differences in evaluation methods, project settings and evaluation contents, resulting in a lack of comparability between studies. More effort should be made to develop more all-sided instruments, which could better help people improve their level of health information literacy and health strategies.

In addition, given the deepening research, study populations have also become more granular, from general populations to disease-specific populations. A positive phenomenon is that the HIL of specific groups started to become the focus. Some studies have started to design suitable evaluation tools for such groups [58,59]. However, such research is rare and not in-depth, and no authoritative evaluation scale has been formed. Given the rapid development of information technology and instantaneous prevalence, people can more easily obtain health information from the internet, which means massive changes in how people receive health information. In a previous study, the HIL of the elderly was described in online health forums and further analyzed to reach relevant conclusions [60]. Different from the traditional ways to obtain information, recognition and utilization of OHI may be the trend in future research. Thus, the ability of people to search, identify and utilize network information services should also be included in the assessment scope.

However, important to note is that many tools were designed from the perspective of expert evaluation and not based on the level of public understanding [61]. Given diverse populations with different social backgrounds and cultural education, the differentiation in understanding greatly influences the accuracy of the evaluation tool [27,28,62]. Therefore, item settings should be carefully considered in future research.

Moreover, besides normal health literacy, public health literacy also started to enter the field of vision of researchers [63]. As the subconcept of health literacy, it was also worth paying attention to the public HIL related to public health literacy. Additionally, the assessment tools covering the conceptual foundations, critical skills and civic orientation in an individual or group could also be investigated.

Each individual’s level of HIL can be directly related to their health condition. The manner adopted by individuals with different levels of HIL influences patient behavior regarding care and health outcomes. The ongoing development of instruments suggests that future research should certainly be aimed at developing comprehensive assessment tools that can systematically measure an individual’s ability to read, seek, understand, evaluate and apply health information on a combination of specific diseases. In addition, from the audience’s perspective, designers should consider diverse factors affecting HIL, such as different educational attainment and comprehensive ability. Moreover, designers need to strike a balance between the assessment tools needed in busy clinical settings and primary healthcare environments.

### 5.2. The Doctor–Patient Relationship

There is no doubt that a gap exists between medical staff and patients in view of medical knowledge [64], which may directly induce the phenomenon that patients tend to believe in medical staff without conditions and lack communication with them. Nevertheless, with the popularization of health information and the population’s awareness of medical participation in recent years, the framework of the doctor–patient relationship is beginning to change swiftly. One of the most significant reasons is the variation in HIL in the population and the diverse approaches to obtaining health information, especially on the internet [65,66].

On the basis of previous studies, good HIL was found to effectively improve the doctor–patient relationship and increase patients’ treatment compliance [67,68]. People with high HIL levels are more willing to communicate with the medical staff to receive disease-related information [69] and proactively join the medical process to make decisions [70,71]. In addition, good HIL helps patients alleviate negative feelings, such as anxiety and depression [44], improve psychological security [72] and raise benign expectations [51,52]. Meanwhile, the doctor’s feedback to the patient also shows the importance of people’s HIL. Good feedback could encourage patients to increase their doctor–patient interactions and enhance their trust [73,74]. Thus, it is necessary to raise people’s HIL level so that the medical process and doctor–patient relationship can be markedly improved.

However, there is still a long way to go before health education is effective. Sometimes there is an explosion of uncensored health information on the internet. As a nonmedical staff, it is difficult for people to identify information that is true or false. When they receive OHI that is not correct, conflicts sometimes arise between doctors and patients. People may argue with the medical staff about their medical content or perform an unauthorized change in medical supervision, which reduces the doctor’s authority [44,75]. In contrast, the doctor’s feedback to the patient may also discourage patients from searching for OHI in the future [76]. Additionally, it has been demonstrated that the outcome of diseases is affected by the doctor–patient relationship [77]. Thus, it is valuable and extremely urgent to help patients receive the right OHI by improving their HIL.

### 5.3. Acquired Platform Media and Quality

Because of its convenience and rich content, OHI has gradually become popular and the main reference for people to use [78]. With so much health information on the internet, people may feel confused about a substantial amount of conflicting information. There is no doubt that the correct OHI exists on the internet, which teaches people not only what to do but also why [79]. Such health information can be effective in helping patients. However, it is inevitable that uncensored and misleading OHI also exists on the internet and harms many people [80]. The health information systems on such platforms are incomplete and not updated in a timely manner. Additionally, more than half of OHI cannot be reviewed before they propagate [4]. Thus, it is necessary for OHI to be evaluated and guided by a professional [81]. Broadening platform media and strengthening vetting also make sense.

Many studies have proven that HIL appears to have a certain social gradient in the population, which is one of the decisive factors related to broad social health problems [46,82,83]. Elderly people, as the key group, have a large demand for HIL [40,84]. Nevertheless, there are fewer approaches for elderly people to utilize [31,32], and some of them are not willing to receive OHI [85]. People tend just to view the first few links obtained from searches using generic search engines, and they do not check the author or owner of the website [86]. Based on this phenomenon and other studies, Crespo J. [87] considered that most users seem focused on quickly finding information rather than evaluating the information they found. Therefore, it may be worth providing a suitable education for them and positively encouraging them to seek health information [28]. In addition, it is necessary to provide older people with more accessible health information. On the basis of the above, such accessibility may be relatively effective in alleviating the influence of social gradient [28].

Finally, online platforms for doctor–patient communication have gradually become popular in recent years [88]. They attract an increasing number of people to receive OHI from medical staff due to their convenience and professionalism. Current studies have paid more attention to offline doctor–patient communication to improve HIL, together with a few studies on online doctor–patient communication. Hence, more studies could be devoted to the latter and create possibilities for medical staff to play a new role in the delivery of health services and education [14].

## 6. Conclusions

The results of this review contribute to summarizing the different assessment tools, influencing factors and current HIL situation. Regarding the number of assessment tools, it is still difficult to comprehensively evaluate the HIL of the population. Some of them may focus only on partial HIL abilities and cannot effectively assess the capacities of cognition, search, evaluation and utilization related to HIL. Moreover, the effect of various disease groups on the assessment results should also be considered. Targeted evaluation tools have certain research potential in assessing and improving the HIL of people. Additionally, it may also be valuable to explore and design assessment tools to investigate public health information literacy.

In this review, we conclude the three different types of influencing factors that affect people’s behaviors and attitudes toward health information from different perspectives. Studies have shown that interest in health information and the demand for information contributes to improving HIL [6,31]. Thus, it is necessary to explore the intrinsic interaction between various influencing factors and people’s attitudes and motivations.

In addition, we preliminarily analyzed the current situations and challenges of HIL. The quality of OHI deeply influences the doctor–patient relationship, while feedback from the medical staff to people induces positive excitation or negative discouragement back to people’s attitude and motivation in seeking OHI. Thus, it is of great importance to reduce the broadcasting of uncensored health information and provide a professional reference for people to improve their HIL. Simultaneously, raising people’s recognition of HIL and cultivating their practical ability to acquire OHI are also crucial aspects of advancing health education for all and improving HIL.

## Figures and Tables

**Table 1 ijerph-20-02706-t001:** Summary of tools assessing health information literacy.

Measurement Scope	Scale Name	Purpose of the Instrument	Scoring	Strengths	Limitations
Screening reading ability of health information	Rapid Estimate of Adult Literacy in Medicine (REALM)	To test an individual ability to read and spell common medical terms and level of literacy in clinical settings	Scores range 0–66: 0–18 = low HIL; 19–45 = medium-low HIL; 45–60 = medium-high HIL; 61–66 = high HIL	It is a rapid and robust assessment tool for administrators.	Only measures one dimension of HIL, without being able to measure the comprehension of health information.
Screening reading and understanding skills of health information	The Test of Functional Health Literacy in Adults (TOFHLA)	An indicator to measure the patient’s ability to read and comprehend health-related materials	Scores range 0–100: 0–59 = inadequate HIL; 60–74 = marginal HIL; 75–100 = adequate HIL	TOFHLA is a valid and reliable instrument in several diverse populations and can be available in different languages.	Completing the whole version of TOFHLA takes a relatively long time.
	The Short Test of Functional Health Literacy in Adults (STOFHLA)	To develop an abbreviated version of the TOFHLA and measure the patient’s ability to read and comprehend health-related materials	Scores range 0–100: 0–53 = low HIL; 54–66 = medium HIL; 67–100 = high HIL	Contrast to long version of the TOFHLA, the short version spares nearly 10 min and has been validated in several diverse populations.	STOFHLA failed to measure other dimensions of HIL, such as the ability to seek, evaluate and apply health-related materials.
	The Newest Vital Sign (NVS)	A screening test for limited literacy in primary healthcare settings	Each item answered correctly is given a score of 1; Score ranges from 0 to 6: 0–1 = inadequate HIL; 2–3 = marginal HIL; 4–6 = adequate HIL	NVS is a rather quick assessment tool for HIL.	The level of HIL might be overestimated due to the small number of entries.
Screening reading and evaluating the quality of health-related materials	The DISCERN questionnaire	Self-report instrument for patients and information providers to judge the quality of written consumer health information	Each item is scored on a 5-point scale ranging from 1 (strongly disagree) to 5 (strongly agree); Higher scores indicate higher HIL levels	The DISCERN questionnaire is a reliable and valid instrument that can be applied to healthcare professionals and patients.	Self-assessment that has the potential for self-report bias.
Proficiency in seeking, understanding and evaluating the quality of health information	Research Readiness Self-Assessment Tool-Health (RRSA-h)	Measure the college-age health information consumers’ proficiency in obtaining, evaluating and understanding of health information	Grade is assigned based on total score that ranges from 0 to 56; High scores = high HIL skills; Low scores = low HIL skills	RRSA-h is suitable for health literacy educators to assess consumers’ skill in electronic health information.	Validation sample did not fully represent a demographically diverse population.
Proficiency in seeking, understanding, evaluation and applying health-related materials	Health Information Literacy questionnaire (HILq)	Evaluate the HIL of patients with chronic kidney disease	Each item is scored on a 5-point scale ranging from 1 (strongly disagree) to 5 (strongly agree); Higher scores indicate higher HIL levels	The HILq for chronic kidney disease patients has good reliability, validity and significant guidance for the health education of CKD.	Self-assessment that has the potential for self-report bias.
Motivation on finding, understanding, evaluation and applying health information	Everyday Health Information Literacy (EHIL)	A practical screening tool to identify individuals with limited EHIL.	Each item is scored on a 5-point scale ranging from 1 (strongly disagree) to 5 (strongly agree); Scores range 24–44: 24–30 = low HIL; 31–33 = medium-low HIL; 34–36 = medium-high HIL; 37–44 = high HIL	The first attempt to systematically assess HIL based on the conceptualization and full of feasibility.	Self-assessment that has the potential for self-report bias.
Motivation on finding, information seeking ability, evaluation and applying health information	Health Information Literacy Self-Rating Scale (HILSS)	A self-rating scale to measure Chinese residents’ HIL	Each item is scored on a 5-point scale ranging from 1 (strongly disagree) to 5 (strongly agree); Higher scores indicate higher HIL levels	A promising tool for wide Chinese residents in primary health care settings to screen for potential information literacy problems.	Self-assessment that has the potential for self-report bias.

## Data Availability

Data is contained within the article. The data presented in this study are available in Table 1.

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
