# Peer review of "Current Situations and Challenges in the Development of Health Information Literacy"

_ijerph, 2023, doi:10.3390/ijerph20032706_

Round 1
Reviewer 1 Report
Although the paper addresses a very important topic and achieves a broad range of different insights into the subject, there is some confusion concerning concepts and presentation of findings (especially in the tools section).
There is another strand of health literacy theory that is not acknowledged but widely recognized internationally: The health literacy concept of Soerensen et al. or the WHO.
E.g., WHO (2013): The Solid Facts Health Literacy or
Sørensen, K., Van den Broucke, S., Fullam, J. et al. Health literacy and public health: A systematic review and integration of definitions and models. BMC Public Health 12, 80 (2012).
Here, there is no distinction made between health literacy and health information literacy which is not at all reflected in this article. It is also not really clear why in the introduction, it is stated that health literacy focuses more on doctor-patient-communication (in contrast to health information literacy) when in the end it is listed as one of the areas of health information literacy.
The assessment tools presented address very different subjects - i.e., assessing the quality of health information on the one hand and assessing peoples' health (information) literacy on the other hand. This is not adequately addressed and is confusing for readers.
There is a discourse on different assessment methods - self-assessment versus assessment of functional health literacy, which is only vaguely referenced. As the main aim of the article is to summarise the tools, concepts and knowledge base on health (information) literacy, this seems to be problematic.
Author Response
Thank you very much for your review. Please see the attachment for reply.

Reviewer 2 Report
1. Some layout of the article should be adjusted.
(1) #79, #88, #105, #115, #124, #135, #143, #158, #192, #220 should be changed to subtitles.
(2) Table 1 is not referenced in this article.2. It is recommended to classify and tabulate according to the scope of assessment tools use.
3. It is recommended to tabulate according to the strengths and weaknesses of the assessment tools.
4. As mentioned in the article, the current assessment tools are not comprehensive. Can you propose specific improvement ideas ?
5. This review is a qualitative study of the current development of Health Information Literacy, with detailed content and a complete structure.
Author Response

(The authors gave the same response as above.)
